# Critical Review of the Benefit from Early Pharmacological and Dietary Support for Patients with Moderate-to-Severe (Non-Terminal) Chronic Kidney Disease

**DOI:** 10.3390/biomedicines13040994

**Published:** 2025-04-19

**Authors:** Charline Danneel, Camille Sauvage, Mohamed Nabil Hayef, Véronique Desmet, Murielle Surquin, Joëlle Nortier, Carine De Vriese

**Affiliations:** 1Department of Pharmacy, Centre Hospitalier Universitaire Brugmann, Place Arthur Van Gehuchten 4, 1020 Brussels, Belgium; mohamednabil.hayef@chu-brugmann.be; 2Department of Dietetics, Centre Hospitalier Universitaire Brugmann, Place Arthur Van Gehuchten 4, 1020 Brussels, Belgium; camille.sauvage@chu-brugmann.be (C.S.); veronique.desmet@chu-brugmann.be (V.D.); 3Department of Geriatrics, Centre Hospitalier Universitaire Brugmann, Place Arthur Van Gehuchten 4, 1020 Brussels, Belgium; murielle.surquin@chu-brugmann.be; 4Department of Nephrology—Dialysis, Centre Hospitalier Universitaire Brugmann, Place Arthur Van Gehuchten 4, 1020 Brussels, Belgium; joelle.nortier@chu-brugmann.be; 5Department of Pharmacotherapy and Pharmaceutics, Faculty of Pharmacy, Université Libre de Bruxelles (ULB), 1050 Brussels, Belgium; carine.de.vriese@ulb.be

**Keywords:** chronic kidney disease, pharmacist, dietitian, patient education, nephroprotection, polypharmacy, interdisciplinary team

## Abstract

Moderate-to-severe chronic kidney disease (CKD) is a public health problem affecting hundreds of millions of people around the world. Started early, nephroprotection measures are able to prevent the degradation of renal function and are a major issue in CKD management. This approach consists of a combination of pharmacological and non-pharmacological treatments aimed at slowing down the decline in renal filtration capacity and improving patient well-being. Drugs such as angiotensin-converting enzyme inhibitors, angiotensin II receptor antagonists, and sodium–glucose cotransport type 2 inhibitors play a crucial role in reducing intraglomerular pressure and renal inflammation. Their beneficial effects are potentiated when they are combined with non-pharmacological approaches, such as salt and protein restriction. This present review provides a critical overview of the current pharmacological and nutritional therapies that may slow down the progression of CKD. Recently, many pharmacological treatments have opened up new perspectives for managing this condition. Nevertheless, prevention remains the cornerstone of effective disease management. Actually, very few studies include both pharmacists and dietitians in their interdisciplinary team mainly represented by nephrologists, nurses, and social workers. However, their specific collaboration may significantly improve the knowledge and skills to help patients in their own CKD management. Future research is required to assess the benefit of collaboration in supporting patients with moderate-to-severe CKD before any concern of renal replacement therapy (RRT).

## 1. Introduction

Kidney Disease: Improving Global Outcomes (KDIGO) defines chronic kidney disease (CKD) as “an abnormality of renal structure or function that has been present for at least three months and has an impact on health”. It is divided into several categories depending on the cause, the estimated glomerular filtration rate (eGFR) according to the CKD-EPI equation (in adults) (stages G1 to G5), and the value of albuminuria (group A1–A3). These three elements make it possible to determine the severity and the risk of progression in a person with CKD [1]. It is often defined as an irreversible and silent disease with little or no somatic symptoms at its onset, but which may be characterized by a progressive impairment of the blood depuration capacities of the kidney, leading to an irreversible functional decrease. It is then represented by an eGFR below 60 mL/min/1.73 m^2^ with or without the presence of markers of kidney damage, such as albuminuria, hematuria, or abnormalities that can be revealed by laboratory or imaging tests [2]. This disease is considered a real public health problem affecting more than 10% of the world population, or more than 850 million people, most of whom live in low- and lower-middle-income countries. This social precarity leads to limited access to diagnosis, prevention, and drug treatment for a large part of the population. In 2017, the global prevalence of CKD varied across regions and populations, reaching around 12% in Europe, 14% in the United States, and up to 16% in Asia. By 2040, it is expected to become the fifth leading cause of deaths worldwide. In addition, the frequently associated co-morbidities (hypertension and type 2 diabetes) precipitate its progression to stage 5, requiring the initiation of renal replacement therapies (RRTs).

Moreover, in 2024, the review by Oosting et al. showed that, in CKD, polypharmacy was associated with an increased risk of therapeutic non-adherence, hospitalization, all-cause mortality, decline in eGFR with decreased quality of life, adverse effects, fractures, and increased risk of cardiovascular disease. Its worldwide prevalence would be in the order of 68% in patients with eGFR between stages 3 and 5, whereas it would be higher when patients switch to RRT with a rate reaching 89% on dialysis and 87% on transplantation [3,4,5].

According to Ibrahim et al., the existing discrepancies between the nephrologist’s drug prescriptions and the daily intake of medications are increased with polypharmacy. More than half of the patients in stages 4–5 had at least one drug divergence and one third had at least two. It is estimated that a non-dialysis patient at stages 3 to 5 in CKD (Figure 1) consumes an average of eight medicines per day whereas a dialysis patient can take a dozen. The cause of these discrepancies is multifactorial: the high number of doctors involved in patient care and an impairment of physical and cognitive abilities [6,7].

On the dietary side, inadequate nutrition can directly or indirectly promote the progression of CKD through obesity, or through its effects on risk factors (diabetes and high blood pressure). Nutritional strategies are important for reducing the risk of complications, such as sodium and volume overload, hyperkalemia, hyperphosphatemia, or even malnutrition, which affect the cardiovascular system, bones, and other organs [9].

Protective measures of renal function should have an essential place in the early stages of the disease. Pharmacotherapy and dietetics are closely linked in the management of CKD. Moreover, the effectiveness of certain medicines may be affected by dietary factors. Excessive salt consumption reduces the effectiveness of renin angiotensin, aldosterone system (RAAS) on proteinuria, blood pressure, and progressive kidney damage. In addition, the combination of a low protein intake with angiotensin-converting enzyme inhibitors (ACEis) significantly reduces proteinuria [4,10].

In this critical review, we focus on the pharmacological and nutritional renoprotective (nephroprotective) measures that have been recently published, from stage 3b (eGFR less than 45 mL/min/1.73 m^2^) to stage 5 without requiring RRT (Figure 1). We also discuss the literature results on the benefits of a pharmacist and a dietitian in the nephrology department.

## 2. Drug and Non-Drug Renoprotection

Drug and non-drug approaches emerged at different times. It was in 2005 that the concept of renoprotection (or nephroprotection) was born, during an experimental observation of the blockage of angiotensin II, by ACEi in chronic proteinuric nephropathies. The researchers have, for the first time, understood that it is not enough to limit themselves to passively accompanying patients to RRT, but that a therapeutic strategy is essential to the preservation of renal function for as long as possible [11]. Nephroprotection therefore aims to slow down the course of CKD while reducing the risk of cardiovascular events [4].

Experimental nutritional interventions emerged in the 1930s, with dietary parameters of varying intensities, in terms of protein and sodium restriction, imposed on rats with CKD. It was from the 1960s that a reduction in protein intake was identified, in particular to reduce uremic symptoms. Moreover, a protein restriction of 30% makes it possible to have favorable effects on sodium intake, which would also be reduced by 28%.

This protein reduction therefore reduces damage by preventing the degradation of renal function and intervenes in the correction of certain complications (hyperphosphatemia, sodium load) [12].

A schematic representation of nephroprotective approaches is shown in Figure 2.

### 2.1. Drug Renoprotection

Any renoprotection involves several drug classes that have demonstrated positive results. The main goals are the control of blood pressure and diabetes mellitus, but also anemia, metabolic acidosis, and phospho-calcic disorders.

#### 2.1.1. Antihypertensive Agents

##### Angiotensin-Converting Enzyme Inhibitors and Angiotensin II Receptor Antagonists

Over the past 20 years, the recommended categories of antihypertensive agents for the strict control of hypertension in diabetic or non-diabetic patients with CKD are ACEis and angiotensin II receptor antagonists (ARAs) with a reduction in the risk of renal failure of 39% and 30% (OR of 0.61 [95% CI, 0.47–0.79] and 0.70 [95% CI, 0.52–0.89]), respectively, compared to the placebo [13]. They are the pillars of the treatment of hypertension from stage 3b. In addition, they are particularly effective in reducing proteinuria by decreasing intra-glomerular filtration pressure [4]. The study by Burnier M. et al. shows that ARAs, in monotherapy or in combination with other antihypertensive agents, effectively reduce blood pressure while improving proteinuria. Indeed, use as monotherapy for 18 weeks to less than 1 year reduces proteinuria by an average of −0.60 g/L (95% CI, 0.93 to 0.26; *p* < 0.01) compared to −1.40 g/L when combined with HCTZ (95% CI, 1.71 to 1.09; *p* < 0.01) [14]. These agents can also induce a significant 16% decrease in progression to end-stage kidney disease (ESKD) [15]. The effectiveness of ACEis and ARAs in reducing intra-glomerular filtration pressure, however, calls for the utmost caution in the event of heart failure and pronounced hypovolemia or dehydration. It is therefore important to monitor biological factors indicating the deterioration of renal function and serum electrolyte disorders (potassium among others) within two-to-four weeks of starting treatment. A steady or severe decrease in more than 30% in eGFR should prompt an investigation for the presence of renovascular disease and discontinuation of treatment. As their effectiveness as monotherapy is limited, it is sometimes recommended to individually adjust blood pressure reduction therapy according to the stage of the disease by combining them with different drug classes to achieve optimal therapeutic results (calcium antagonist, loop diuretic, beta-blocker, alpha-blocker, etc.) [1,4,16,17]. This was particularly the case in the cohort of Muntner et al., where 60% of patients were treated with at least three antihypertensives [18]. Figure 3 illustrates the possible drug combinations from stages 1 to 5 of non-dialysis CKD.

##### Mineralocorticoid Receptor Antagonists

Mineralocorticoid receptor antagonists include steroidal antagonists, such as spironolactone and eplerenone, which effectively reduce blood pressure with a risk of hyperkalemia, and selective non-steroidal finerenone-type antagonists [19]. Spironolactone not only has a positive effect on systolic blood pressure by reducing it by 6 mmHg, but also a positive effect on reducing proteinuria in combination with ACEis and ARAs [20]. Finerenone is the last nephroprotection agent to be approved by the Food and Drug Administration (FDA) in the United States in 2021 for the treatment of CKD associated with type 2 diabetes. The FIDELIO-DKD study demonstrated that patients with an eGFR between 25 and 60 mL/min/1.73 m^2^ had a significant reduction in primary and cardiovascular renal objectives in the treatment of diabetic nephropathy, by reducing the fibrotic and inflammatory processes associated with an overactivation of the mineralocorticoid receptor. A second study (FIGARO-DKD) on patients with less-severe CKD also showed that finerenone offered cardiovascular benefits in type 2 diabetes-related CKD with a significant reduction in renal decline [21,22,23]. This molecule therefore demonstrates therapeutic efficacy not only in hypertensive patients, but also in diabetic patients with microalbuminuria. It is the first molecule available on the market for the treatment of CKD with albuminuria in adults with type 2 diabetes [22]. Therefore, it is recommended for patients with CKD associated with T2D and moderate (A2) or severe (A3) albuminuria if their eGFR is at least 25 mL/min/1.73 m^2^ and their serum potassium is below 5.0 mmol/L [24,25].

However, control of serum potassium remains more than necessary, even though finerenone caused hyperkalemia less frequently than spironolactone or eplerenone (so-called potassium-sparing diuretics) [22].

#### 2.1.2. Antidiabetic Drugs

##### Inhibitors of Sodium–Glucose Type II Cotransport

Numerous studies have shown that certain drugs initially developed to treat other diseases could be potential treatments for CKD. This is the case for sodium–glucose cotransport type 2 inhibitors (iSGLT2) or gliflozins, which were initially used to reduce blood sugar levels in diabetic patients. They actually reduce the risk of kidney function decline and cardiovascular events compared to the placebo. However, in early studies, only 7 to 26% of participants with an eGFR below 60 mL/min/1.73 m^2^ were included. This therefore made it impossible to determine the benefits of treatment for CKD patients. Subsequently, recent studies have rapidly established a nephroprotective effect. Canagliflozin slows down the progression of CKD in diabetic patients, as shown by the CREDENCE study, while dapagliflozin has similar effects on diabetic or non-diabetic patients with CKD, as seen in the DAPA-CKD study. More recently, the EMPA-KIDNEY trial showed that treatment with empagliflozin resulted in a lower risk of progression of kidney failure or cardiovascular death than the placebo [21]. These drugs therefore represent a promising therapeutic option aim at reducing renal workload by decreasing the proximal tubular reabsorption of sodium, leading to a reduction in blood pressure and albuminuria. Two of the three drugs are reimbursed in Belgium and can be started in patients up to an eGFR of 25 mL/min/1.73 m^2^ (for dapagliflozin) or 20 mL/min/1.73 m^2^ (for empagliflozin), with or without albuminuria and type 2 diabetes. Their continued use below this threshold until dialysis or renal transplantation is possible, provided that they are well-tolerated while showing renal and cardiovascular benefits [4]. These iSGLT2s can reduce the risk of progression of kidney disease by 37% (relative risk: 0.63, 95% CI 0.58–0.69) regardless of diabetes [26]. However, a decline in renal function occurs upon their initiation, without the long-term restoration of the initial eGFR. Dapagliflozin is the molecule that presents the weakest profile in this regard [27].

Nowadays, it is clear that there are additive and synergistic effects in the combination of an ARA with an iSGLT2 in terms of reducing patients’ cardiovascular risks. Together, they can reduce intraglomerular pressure by their respective effects on afferent and efferent arteriolar tones. This is why nephrologists are prompted to prescribe these medications [4,28].

##### Glucagon-like-Peptide-1

Glucagon-like-peptide-1 (GLP-1) was originally developed as a hypoglycemic drug. The analysis of the FLOW study describes the benefits of the GLP-1 analog in patients with type 2 diabetes and impaired renal function [29]. This renal protection may be partially due to the fact that treatment improves many cardiorenal risk factors, including weight loss, better glycemic control, lower blood pressure, and reduced dyslipidemia [30].

Semaglutide demonstrated positive effects in reducing the risk of clinically important renal events and cardiovascular death in patients with type 2 diabetes and CKD [29]. Indeed, patients had a 24% lower risk of renal events (HR 0.76, 95% CI 0.66–0.88, *p* = 0.0003) and a slower decline in the eGFR slope of 1.16 mL/min/year (95% CI 0.86–1.47). This protective effect was not influenced by the eGFR and was consistent across all age groups (HR 0.71, 0.85, and 0.63 for patients under 65, 65–74, and 75 years and older, respectively). This new agent is in strong competition with iSGLT2, but the results of the PRECIDENTD study, which ends in 2029, could be informative. It is recommended in cases of CKD to use iSGLT2, which allows for a better cardiovascular risk reduction than GLP-1. Patients with a normal eGFR should be treated with GLP-1, which provides better cardiovascular protection than iSGLT2 [31].

#### 2.1.3. Sodium Bicarbonate Supplementation

The disorder of an acid-base imbalance commonly develops in CKD, as the kidneys play a major role in maintaining acid-base balance. This disorder is proportional to the progression of the disease, which can lead to adverse consequences, such as bone demineralization, loss of muscle mass, or worsening of renal dysfunction [23]. In 2020, it is estimated that 12–20% of US patients with CKD stage G3b and 27–38% of patients with stage G4–5 CKD have metabolic acidosis [32]. The KDIGO 2024 guidelines emphasize the need to consider pharmacological treatment, possibly in combination with dietary changes, to prevent the development of acidosis with potential clinical consequences, such as a serum bicarbonate levels below 18 mmol/L in adults [1].

According to a meta-analysis by Yang T-Y et al., sodium bicarbonate supplementation may be beneficial in preventing the deterioration of renal function while increasing muscle mass. However, the treatment was also associated with a statistically significant increase in serum calcium and phosphate. It may also be associated with higher BP; however, more extensive research is needed to confirm these results [33].

#### 2.1.4. Erythropoiesis-Stimulating Agents

In the early stages of CKD, hemoglobin levels are generally maintained. However, they begin to decrease when the eGFR usually drops below 45 mL/min/1.73 m^2^ leading to tissue hypoxia, which contributes to further progression. There are three distinct strands of care. The first is to exclude other possible causes of anemia (inflammation, blood loss, hemolysis, hematological disorders, vitamin B9/B12 deficiency). The second is to diagnose and treat iron deficiency. Finally, if anemia persists, it is recommended to use an erythropoiesis-stimulating agent [34,35].

#### 2.1.5. Phosphorus Binders

Abnormally high phosphorus levels in CKD can lead to bone disease, vascular calcification, and cardiovascular disease. To combat this, phosphorus binders are available. They bind to phosphorus in the gastrointestinal tract to form an insoluble complex, reducing its absorption and preventing long-term complications. The effectiveness of these binders’ hinges on their binding strength. On the market, binders can be classified into two groups: calcic and non-calcic. The selection of a chelating agent is determined by the individual’s medical history. The 2017 KDIGO guidelines recommended the limited use of calcium binders (calcium carbonate) to avoid the development of vascular calcification, hypercalcemia, or adynamic bones. In case of these complications, a switch of a non-calcium binder (sevelamer, lanthanum carbonate) is efficient [1,36,37].

### 2.2. Non-Drug Renoprotection

In addition to the arsenal of pharmacological products, the prevention of CKD is also based on the correction of risk factors by hygienic–dietary measures. They start with smoking cessation and weight management, then extend to physical activity and a diet adapted to the stage of the disease. A regular assessment of risk factors should be carried out every 3 to 6 months [1].

#### 2.2.1. Lifestyle Measures

Since smoking is considered as risk factor responsible for worsening kidney disease, it is recommended to discuss smoking cessation with patients to prevent further vascular-type organ damage [38,39,40]. With regard to physical activity, KDIGO recommends regular and moderate physical activity of at least 150 min per week or equivalent to the patient’s cardiovascular or physical tolerance. The latter has many cardiovascular, mental, and social benefits for the patient [1].

Weight gain is associated with a higher risk of CKD progression. A decrease in proteinuria and blood pressure is observed in the case of weight loss following a change in eating habits or the resumption of physical activity. Therefore, weight loss is recommended in obese patients, thus allowing a likely slowdown in the decline in the eGFR [41].

#### 2.2.2. Dietary Strategies

Dietary strategies rely on specific therapeutic feeding to curb the decline in renal function, in particular protein and salt restriction, as well as the control of potassium and phosphorus food content.

##### Protein Intake Restriction

Excessive protein intake leads to several adverse effects on the kidneys, such as increased renal blood flow, intraglomerular pressure, and renal hyperfiltration, consequently favoring an increased workload of the kidneys with damage to the glomerular structure, and therefore an impact on the progression of the CKD [42]. In this meta-analysis, reduced protein intake was shown to decrease the risk of disease development (OR: 0.59, 95% CI: 0.41 to 0.85), as well as the decline in the eGFr by 1.85 mL/min/1.73 m^2^ (95% CI: 0.77 to 2.93, *p* = 0.001). Decreased urinary protein was significantly decrease by 0.44 g/day (95% CI: −0.80 to −0.08, *p* = 0.02) compared to the control groups [43]. This excess in dietary protein may also be linked to hyperphosphatemia as well as metabolic acidosis [44]. The recommendations advise a protein intake range of 0.55 to 0.60 g/kg body weight/day or 0.28 g to 0.43 g/kg body weight/day combined with keto analog supplementation. However, its implementation requires patients’ motivation and a rigorous follow-up. In practice, it seems more feasible to target protein intake levels between 0.6 and 0.8 g/kg body weight/day in stage 3b [45,46,47,48].

The current data are still insufficient to advocate a specific type of protein and do not provide a consensus. However, plant-based proteins appear to have nephroprotective effects on proteinuria, disease progression, circulating uremic toxins, phosphorus intake, and endogenous acid production [42]. Wathanavasin W. et al. also reported that an increase of 10 g of vegetable proteins was linked to a decrease in annual eGFRs, and that the intake of proteins of animal origin showed no significant link to a decrease in eGFRs (0.01; 95% CI, −0.04 to 0.05 mL/min/1.73 m^2^ for an increase of 10 g of animal proteins) [49]. Some studies highlight the beneficial effects of plant-based protein intake on various renal and cardiovascular aspects [47].

##### Sodium Chloride Intake Restriction

The restriction of sodium chloride in CKD is one of the essential practices following the inability of the kidneys to excrete this sodium [50]. Reducing salt intake helps regulate blood pressure, among other things, by reducing both systolic and diastolic blood pressure, from the early stages of CKD, as demonstrated in this meta-analysis (6.91/−3.91 mm Hg (95% CI −8.82 to −4.99/−4.80 to −3.02) [51].

Reducing salt intake helps, among other things, to regulate blood pressure and proteinuria, to reduce the risk of kidney damage and disease progression, and to reduce the development of cardiovascular disease [46,52]. According to the study by McMahon M. et al., high sodium consumption is associated with a higher risk of CKD (OR, 1.21; 95% CI, 1.06 to 1.38). Moreover, from the early stages of CKD, reducing one’s salt intake can reduce sodium excretion (MD −83.81 mmol/day, 95% CI −104.54 to −63.08) [51,53].

A reduction in dietary sodium intake allows a significant improvement in proteinuria, of 0.4 g/day (95% CI: 0.2–0.6 g/day). In addition, this reduced intake improves the effect of certain drugs, such as ACEis and ARAs. This reduction helps increase the antiproteinuric effect of RAAS blockers [10,54]. To this end, KDIGO recommends a sodium intake of less than 2000 mg per day, the equivalent of 5 g of sodium chloride per day [1].

##### Potassium Intake Restriction

Patients with CKD are at greater risk of developing electrolyte disorders, including hyperkalemia [1,45]. Constipation may also be associated with a deterioration in kidney function due to several factors, including low dietary fiber intake, usually related to dietary potassium restriction. However, these fibers play an essential role in maintaining the balance of the gut microbiota, which in turn plays a key role in reducing uremic toxins. Moreover, thanks to the richness in dietary fiber in plant-based diets, gastrointestinal transit is improved and favors increased fecal excretion of potassium and reduced constipation [49]. According to Wathanavasin W. et al., dietary fiber supplementation ranging from 6 to 50 g daily over a 4-week period could significantly decrease serum levels of uremic toxins, including P-cresyl sulfate, indoxyl sulfate, and urea nitrogen (SMD −0.22, −0.34, and −0.25, respectively, with *p* values < 0.05) [55]. This type of diet would not be associated with hyperkalemia for patients with CKD stages 3 and 4 [49]. Mahboobi S. et al. also report that serum potassium levels remain unchanged despite increased fruit and vegetable intake in a very-low-protein vegetarian diet. These results suggest that a plant-based diet may be more effective in preserving eGFRs compared to a diet with a low plant intake [56].

The meta-analysis of Gai W. et al. demonstrated a reduction in all-cause mortality, cardiovascular mortality, and cardiovascular event risk via elevated dietary fiber intake in patients with CKD. An association was made between higher dietary fiber intake and lower all-cause mortality (HR 0.80; 95% CI 0.74–0.86, *p* < 0.001). Also, higher dietary fiber intake was related to lower cardiovascular mortality (HR 0.78; 95% CI 0.67–0.90, *p* < 0.001). Moreover, low dietary fiber intake was associated with an elevated risk of cardiovascular diseases (HR 0.87; 95% CI 0.80–0.95, *p* < 0.05) [57]. The meta-analysis by Kelly M. et al. demonstrates that increased vegetable consumption and higher potassium intake were associated with lower risks of CKD [53].

Dietary management in the event of hyperkalemia is based on individual dietary advice based in particular on the bioavailability of potassium in food (vegetables and fruits) and on certain cooking methods to be preferred, such as cooking with water, or soaking plants (unlike other dry-cooking or dehydration methods) to remove potassium from certain foods [4,44,58]. Boiling and soaking can reduce the potassium content of some foods by 60 to 80% [49].

The so-called Mediterranean diet, which promotes the intake of dietary fiber with fruits, vegetables, whole grains, and unsaturated fats, has shown several beneficial effects in terms of controlling high blood pressure, dyslipidemia, systemic inflammation, metabolic syndrome, diabetes, body weight, and obesity. The Mediterranean diet seems to have a positive impact at different stages of CKD, contributing to its prevention while helping to reduce its progression to more advanced stages [45,59]. Indeed, this type of diet is associated with better kidney function in a population of elderly men. Adherence to it would be linked to a better survival rate in CKD patients [60]. The Mediterranean diet is associated with a slower decline in kidney function and progression of CKD. It also improves lipid profiles, glucose metabolism, and blood pressure, reducing the risk of developing cardiovascular disease in patients with CKD. Finally, this type of diet is associated with a reduced risk of mortality in patients with CKD [35].

Moreover, the benefits of a plant-based diet include a decreased production of uremic toxins, an improvement in intestinal dysbiosis, a reduction in phosphorus load, a decrease in glomerular hyperfiltration, the prevention of vascular calcification thanks to an increased intake of magnesium, as well as a reduction in cardiovascular risk and a slowing down in the progression of CKD [42,47].

##### Phosphorus Intake Restriction

Hyperphosphatemia develops with the progression of CKD and leads to complications, such as secondary hyperthyroidism and disorders of bone metabolism, and is also a common complication leading to an increased risk of cardiovascular morbidity–mortality and vascular calcification [1,61].

Due to the presence of phytates, bioavailability is reduced when phosphorus comes from plants (20 to 40%) compared to that from animal proteins (40–60%). The latter is itself less bioavailable than phosphorus of additive origin and processed foods (100%). Dietary adaptations apply in case of hyperphosphatemia, the aim being to find serum levels in the standard range [48,62]. According to Mahboobi S. et al., dietary interventions reduced serum phosphate levels and increased serum calcium levels compared with the control group (DM −1.22 (95% CI −2.34, −0.10)) and (DM 0.51 (95% CI 0.30, 0.73)) [56].

##### Hydration Recommendations

Until stage 3 of CKD, urinary flow is generally preserved [63]. According to Ákos Géza Pethő et al., the daily amount of fluid without increasing body weight is achieved with a consumption of between 2 and 2.5 L per day [35]. When fluid intake is reduced, waste must be concentrated by the kidneys, which results in an overload of waste excretion in the urine (1 L instead of 3 L). However, fluid overload should be avoided [35].

The study suggests a daily water consumption value of 1 to 2 L per day for CKD patients. A high water intake helps prevent the onset of CKD, but specifies that higher consumption is not useful for moderate and advanced stages of CKD. Moreover, a lower prevalence and slower progression of kidney function decline have been shown [64].

However, a high fluid intake is not necessarily associated with a reduction in long-term cardiovascular mortality [65]. Along these lines, controversies still exist in terms of the adequate volume recommended to CKD patients. It remains evident that an individualized assessment of the hydration status of each patient is required.

## 3. Nephrotoxic Drugs

The patient’s knowledge of nephrotoxic drugs is intended to make them aware of the harmful consequences of certain drugs on their kidneys. Exposure to anti-inflammatory nonsteroidal drugs can lead to adverse events, such as acute kidney damage due to excessive vasoconstriction of the afferent glomerular arteriole, more emergency room visits and hospitalizations, and a higher risk of mortality. Sodium phosphate solutions used in a colonoscopy are also strongly discouraged because they can lead to the intrarenal precipitation of phosphate crystals [66,67]. Finally, the administration of iodized contrast agents should be advised and considered with a preference for non-ionic products with fewer adverse effects [68]. Laville et al. demonstrated in 2018 that the proportion of inappropriate prescriptions in CKD increases with the severity of renal function. In stages 4 and 5, 57% of patients had at least one contraindicated medicine, and 42% received a prescription where the dose was inappropriate (Figure 4) [69].

## 4. The Interdisciplinary Team in Nephrology

### 4.1. Background

Historically, CKD management has primarily focused on dialysis therapies as the most frequent RRT. Nowadays, the focus has been on the earlier stages of the disease. The REVEAL-CKD study clearly demonstrates that patients and clinicians should engage in kidney health management by informing patients about kidney function tests, as this could help them become actively involved in decisions about their own healthcare [70]. As a result, an interdisciplinary care team was proposed, composed of healthcare professionals specialized in this field. It is therefore recommended that the interdisciplinary team takes an active part in supporting patients from the early stages, to delay the progression of the disease and manage complications.

The KDIGO 2012 guidelines state that an interdisciplinary team should have access to dietary advice, as well as providing information on all aspects of dialysis, including treatment options, the creation of arteriovenous access, as well as ethical, psychological, and social care services. This team should consist of nephrologists, nephrology nurses, psychologists, social workers, and renal dietitians [71]. However, some aspects were insufficiently addressed, including the appropriate use of medicines, the monitoring and management of pharmacotherapy, as well as the follow-up of drug problems [72]. Of interest, the KDIGO 2024 awareness campaign proposed the clinical pharmacist to join the interdisciplinary team [1].

In Belgium, the National Invalidated Health Insurance Institute (INAMI) formalized, in July 2024, a new refund in nephrology concerning the interdisciplinary assessment of CKD patients with a eGFR < 20 mL/min/1.73 m^2^. The dietitian is included in the interdisciplinary group by the care path. Unfortunately, the pharmacist is not mentioned [73]. The World Health Organization (WHO) defines therapeutic patient education as an educational activity that focuses on the patient and/or their caregivers. This approach is crucial for individuals living with chronic illnesses, as it empowers them to make informed decisions and enhances the efficiency of the healthcare system. In the context of CKD, therapeutic patient education is a key component of interdisciplinary care [74]. In the field of CKD, it is one of the essential pillars of interdisciplinary care. Indeed, education is an individualized pedagogical project, including different notions, such as strengthening knowledge, skills, and confidence, and allows us to meet the needs of patients and improve their quality of life while coping with the disease. These strategies involve focusing on drug recommendations, maintaining regular physical activity, and adopting an appropriate diet [75,76,77]. Ultimately, the aim is for the patient to become a partner in his or her own care.

Several studies have shown that the effectiveness of certain drugs is increased by dietary factors (Figure 5). First, reduced salt intake improves the effect of certain drugs, such as ACEis and ARAs, while reinforcing the nephroprotective properties of their inhibition [2,10,78,79]. Moreover, a daily salt intake of 14 g inhibits the anti-proteinuric effect of ACE i, leading to an increase in the progression of kidney decline [10]. Low-protein diets may have synergistic effects when used in conjunction with renin–angiotensin–aldosterone and iSGLT2 to mitigate proteinuria and CKD progression.

Kalantar-Zadeh et al. put forward the hypothesis that dietary protein reduction could further improve the properties of iSGLT2 on albuminuria [79]. Monitoring of nutritional status and body composition in CKD patients consuming iSGLT2 is recommended, as they may contribute to weight loss, specifically the loss of lean mass. This loss could be harmful to elderly patients [78].

Moreover, several controlled trials have shown a reduction in the degree of albuminuria of 10 to 20% thanks to a partial replacement of animal proteins with those of plant origin, with a combination of angiotensin-converting enzyme inhibitors [49].

### 4.2. Place of Clinical Pharmacists and Dietitians in Early CKD

Many studies demonstrate the role of the pharmacist at the bedside of hemodialysis patients, but the implementation of optimal medication management in primary care is crucial to ensure long-term benefits in the fight against the progression of CKD. To achieve this, pharmacists should be members of the interdisciplinary team, in collaboration with nephrologists, for the optimization of medications and the improvement of the quality of care. They ensure that patients adhere to their treatment while identifying inappropriate medicines that need to be discontinued or those that need to be rehabilitated based on patients’ eGFRs [40,80]. They are best-trained in drug management by intervening in the in- and/or out-of-hospital comparative assessments as well as in disease education and advice on controlling risk factors in self-management (Table 1) [81,82]. The reasons are multifactorial: CKD patients are polymedicated, and those at risk of medication-related problems have an even greater vulnerability to drug poisoning and increased risk of drug interactions. Consequently, the pharmacist will be useful to adjust drug doses in the case of renal failure [83]. To this end, they use educational tools based on visual learning and simple vocabulary to facilitate communication with patients and improve the effectiveness of drug advice [84].

The aim is to improve the quality of care while assessing the efficacy and toxicity of treatments according to the medical indications [1,82]. However, their services are under-exploited to improve patient adherence to treatment. This situation is due, among other things, to the lack of time devoted to implementing interventions aimed at improving adherence, and the limited number of pharmacists working in these areas [84].

The study by Onor I et al. demonstrated that the interventions performed by clinical pharmacist led to significant improvements in clinical (reduction in hospitalization rates), economic (reduction in direct and indirect costs), and human (improved well-being) outcomes in CKD patients [82]. Indeed, systematically integrating clinical pharmacists in the interdisciplinary team for CKD patients could help achieve the goals for blood pressure, blood sugar, but also the early assessment and treatment of proteinuria, anemia, and secondary hyperparathyroidism [72]. According to Lin E et al., the treatment of CKD requires effective management on several levels, including blood pressure control, reduction in cardiovascular risk factors, diet management, and water and electrolyte balance. In pre-dialysis CKD patients Seng J. et al., highlighted three types of factors associated with poor therapeutic adherence: (1) patient-related factors, including misconceptions about medications and a lack of perceived self-efficacy when it comes to taking them; (2) treatment-related factors, including polypharmacy and a loss of trust in the doctor; and (3) socioeconomic factors, including low social support and low levels of education [85]. It emphasizes the importance of appropriate interventions by renal healthcare professionals, particularly clinical pharmacists, to improve medication adherence in this high-risk patient group [86].

The dietitian specializing in the field of nephrology has a predominant and essential role in preventing, diagnosing, and treating malnutrition while playing a role in educating patients. Indeed, it accompanies patients in adapting their eating habits (sodium, phosphorus, potassium, and protein intake) according to their individual needs, the severity of their CKD, and the comorbidities [1]. Nevertheless, its role goes far beyond explaining dietary restrictions. Indeed, the dietitian must arouse the interest of patients to change their eating habits, while ensuring a good understanding of the explanations provided [87]. Moreover, it is recognized that its role is essential in the education of food adapted to the CKD, with personalized dietary and nutritional advice [88].

However, CKD prevention relies on dietary education emphasizing the importance of a healthy diet. The dietitian therefore has an essential role in the early stages of CKD [89]. Indeed, to prevent and reduce the risk of CKD, adherence to a healthy diet is recommended because interventions aimed at promoting healthy eating behaviors could reduce the incidence of CKD [90,91]. The complexity of dietary prescriptions and dietary restrictions leads to pre-existing dietary and lifestyle changes, which can negatively impact dietary adherence in patients with CKD. Only 31.5% to 68.5% of patients adhered to dietary advice according to some studies. Adherence to a specific diet is not necessarily linked to patients’ nutritional knowledge. Indeed, other factors come into play, such as biological, personal, cognitive, environmental, and sociocultural dimensions [92].

We also know that dietary habits are involved in the pathogenesis and progression of CKD, particularly in cardiovascular and metabolic risk factors, such as high blood pressure, but also through increased consumption of certain foods [90]. Dietary management no longer focuses on specific, isolated nutrients, but rather on a holistic approach that takes into account patients’ eating habits [63,90]. These would facilitate the referral of patients to nutritional advice based on different healthy food groups. Adopting healthy eating habits would reduce mortality by 27%, although this did not show significant effects on ESKD [93]. The effectiveness of this support depends on the underlying pathology but also on the patient’s commitment. Dietary and lifestyle changes complement drug treatment. To increase the health benefits of CKD patients, access to information and resources is necessary to adopt a therapeutic diet adapted to the specific needs of patients, depending on their stage of CKD [35].

### 4.3. Benefit in Increasing Patient Adherence to a Combined Pharmacological and Dietary Approach

Adherence to pharmacological and dietary treatments is crucial for treatment success, but is often difficult to achieve. Therefore, setting up an interdisciplinary program could delay the need for RTT, extend life expectancy, and meet cost-effectiveness criteria in mild or moderate CKD. However, the pharmacist is rarely associated with the dietitian’s work, although recent research shows its usefulness [94]. Indeed, in Cho et al.’s study, patients who received interdisciplinary training with a nephrologist, dialysis nurse, pharmacist, dietitian, and social worker before dialysis achieved better results, with a reduction in unplanned urgent dialysis, cardiovascular events, and hospital days, compared to those who did not have access to this training [95].

In the study by Donald M et al., where the drug and diet aspects were integrated into the self-management of the disease, the authors observed that the participants wanted to develop their knowledge about drug interactions and adverse effects of certain treatments, but also about their ability to adapt their diet through personalized advice to help them better manage their CKD [77].

Johns et al. showed that interdisciplinary teams comprising a clinical pharmacist and a dietitian have a positive effect on patient education while preparing patients for end-stage CKD, which is associated with better clinical outcomes [40]. Pehlivanli et al. also showed this in their study by explaining that it would be wise to call upon a clinical pharmacist, alongside doctors, nurses, nutritionists, and other health professionals because they can contribute to therapeutic monitoring by preventing drug-related problems (DRPs) likely to occur during the care of MRC patients [96].

## 5. Conclusions and Perspectives

Drug and non-drug nephroprotection are major issues in the management of CKD. In drug strategies, some treatments slow progression and others target cardiovascular risk factors to prevent complications. However, these interventions need to be coupled with non-drug approaches, such as protein and sodium restriction, but also lifestyle adaptation (smoking cessation and regular physical activity). Combining these approaches, rigorous clinical follow-up, and appropriate therapeutic education are essential to optimize patient support, improve quality of life, and delay disease progression.

In our institution, we plan to carry out a study to assess the effectiveness of the collaborative approach of the pharmacist as an expert in drug treatments and the dietitian as a specialist in food and nutrition in order to better understand the impact of this pair on patients’ clinical outcomes, while underlining the importance of comprehensive and interdisciplinary management in the context of CKD. By working closely together, these two fields could improve drug and food adhesion, and therefore slow down the progression of CKD. In parallel, a study on the development of an education program could shed essential light on the complementarity of these two professions.

## Figures and Tables

**Figure 1 biomedicines-13-00994-f001:**
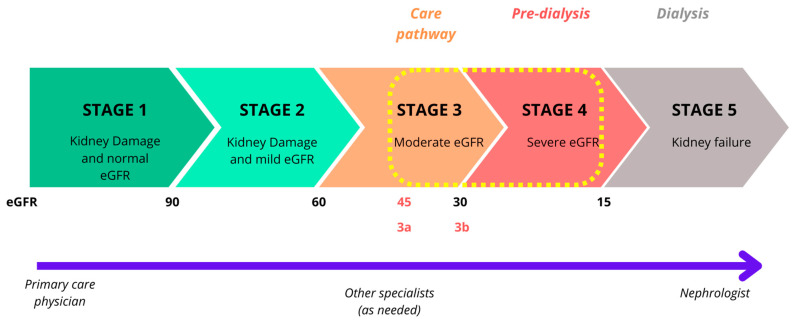
The 5 stages of chronic kidney disease. Adapted from [8]. Abbreviations: eGFR, estimated glomerular filtration rate.

**Figure 2 biomedicines-13-00994-f002:**
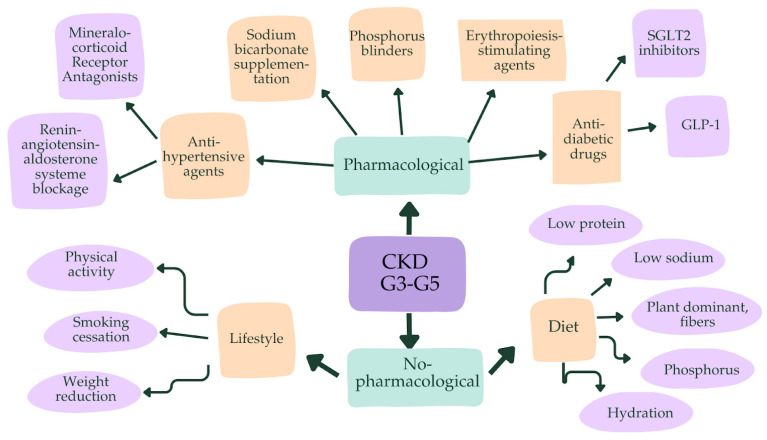
Summary of renoprotective approaches. Abbreviations: CKD, chronic kidney disease; GLP-1, glucagon-like-peptide 1; SGLT2, sodium–glucose transport protein 2.

**Figure 3 biomedicines-13-00994-f003:**
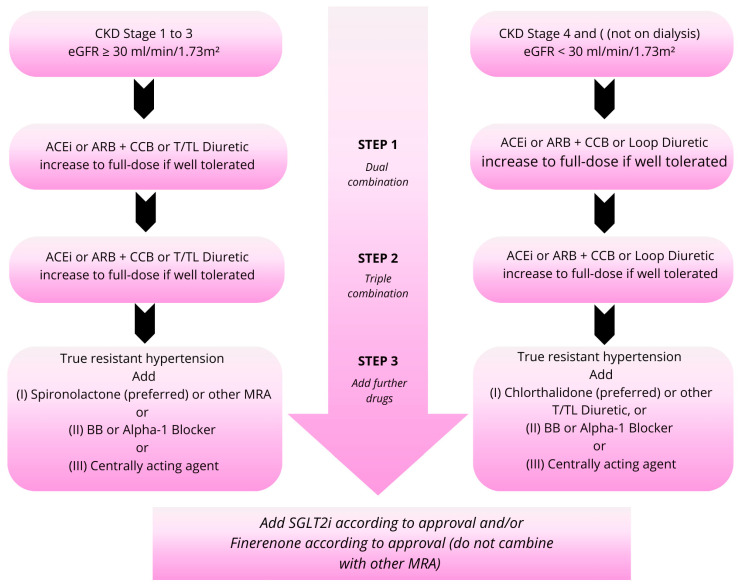
Recommended hypotensive treatments depending on the stage of CKD. Adapted from KDIGO 2024 [16]. Abbreviations: ACEis, angiotensin converting enzyme inhibitors; ARBs, angiotensin II receptor blockers; BB, beta blocker; CCB, calcium channel blocker; CKD, chronic kidney disease; eGFR, estimated glomerular filtration rate; T diuretic, Thiazide diuretic; TL diuretic, Thiazide-like diuretic; SGLT2i, sodium–glucose cotransport type 2 inhibitor.

**Figure 4 biomedicines-13-00994-f004:**
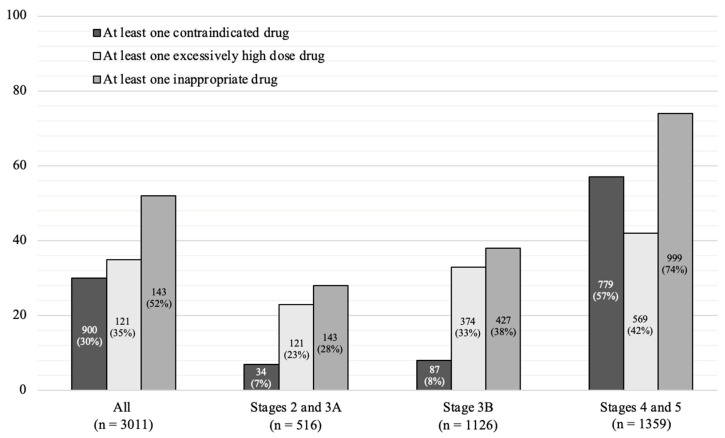
Percentage of patients by CKD stage with at least one contraindicated, inappropriate high-dose or inappropriate prescription. Adapted from [69].

**Figure 5 biomedicines-13-00994-f005:**
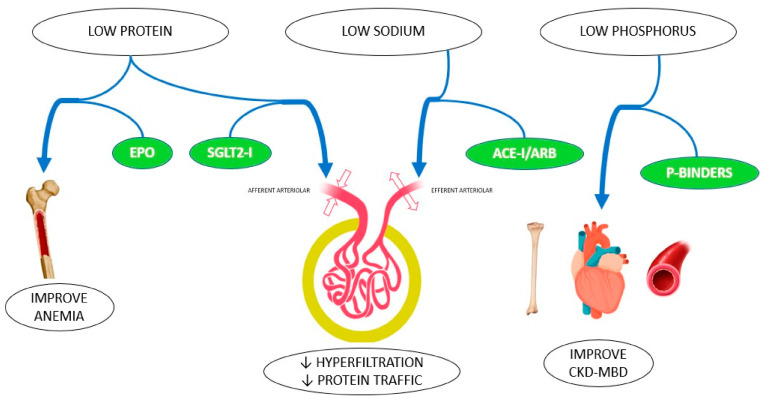
Synergy between nutritional intervention and pharmacological treatment in CKD patients [10]. Abbreviations: ACE-I, angiotensin-converting enzyme inhibitor; ARBs, angiotensin II receptor blockers; EPO, erythropoietin; P-binders, phosphorus binders; SGLT2i, sodium–glucose cotransport type 2 inhibitor.

**Table 1 biomedicines-13-00994-t001:** The step-by-step process of overall medication management from the early stage of CKD by the clinical pharmacist.

** *Comparative Review of Medicinal Products* **	- Determine the actual medication list - Resolve discrepancies between the actual medication list and the one in the medical record
** *Drug Management* **	- Identify and resolve medication-related problems (drug interactions, adverse reactions, indications, dose, route of administration, efficacy, and safety) - Assess therapeutic adhesion and causes of non-adhesion - Educational advice to the patient
** *Deprescription* **	- Identify and stop inappropriate medications in collaboration with healthcare providers.
** *Medication Agreement* **	- Communication with other doctors

## Data Availability

Not applicable.

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
