# Peer review of "Critical Review of the Benefit from Early Pharmacological and Dietary Support for Patients with Moderate-to-Severe (Non-Terminal) Chronic Kidney Disease"

_biomedicines, 2025, doi:10.3390/biomedicines13040994_

Round 1

Reviewer 1 Report

Comments and Suggestions for Authors

The article presents a kind of state of the art and aims to highlight the role of the interdisciplinary team in the care of patients with chronic kidney disease (CKD). The topic is interesting, plus, emphasizing the role of newly introduced team members may engage the readers. However, the goal of the paper is not clearly defined.

The title lacks information that this is an overview article. Moreover, the paper includes a lot of very basic information, such as the items related to diet in CKD. If so decided, it is not clear why other are lacking, for example, references to fluid balance. There are also concerns about the references being often presented at the end of a paragraph in a bulk manner, e.g., line 133 or line 330. At the same time, citations are missing in many places (e.g., line 309-311: The KDIGO 2012 guidelines explained that interdisciplinary nephrological care should take into account patient education about dietary advice, renal replacement modalities, and transplantation [?]; or Line 175-177: It could become a drug of choice in CKD, compared to steroidal mineralocorticoid receptor antagonists [(?]).

The abstract presents an interdisciplinary team consisting of a medical doctor, nurse, and social worker. The roles of nurse and social worker are not articulated in the paper, especially in the context of pharmacological and dietary support for patients with CKD. Moreover, in lines 311-312, it states, “This team should consist of nephrologists, nephrology nurses, and renal dieticians;” thus, the social worker is not mentioned. It is not explained in any way.

The article contains many inconsistencies. For example, Chapter 2.1 is dedicated to Drug Nephroprotection, and its sub-chapters are focused on medications (2.1.1 - 2.1.3). So, why is Chapter 2.1.4 dedicated to Metabolic Acidosis and Chapter 2.1.5 to Anemia, while these are not drug categories. Chapter 2 is titled Drug and Non-drug Nephroprotection, so it is unclear why sub-chapter 2.3 is dedicated to Nephrotoxic Drugs. In Chapter 3 (The Interdisciplinary Team in Nephrology), there is a sub-chapter titled The Place of Clinical Pharmacist and Dietitians in CKD, but it contains only one paragraph, the middle one, focused on the dietician. This chapter seems to be the essence of the paper, but the roles of other team members are not adequately presented.

In conclusion, the article must be principally reconsidered before being accepted for publication. This includes removing basic textbook information, clearly defining the objective, and consistently presenting it throughout the paper.

Author Response

Comments 1: The title lacks information that this is an overview article.

Response 1: Thank you for pointing this out. I agree with this comment. We have changed it in the manuscript, page 1, line 1.

Comments 2 :the references being often presented at the end of a paragraph in a bulk manner, e.g., line 133 or line 330.

Response 2: We agree, so we have amended the sources on page 4, paragraph: 2.1.1.1. Angiotensin-converting enzyme inhibitors and angiotensin II receptor antagonists. For line 330, we have reviewed the sources and believe that they correspond to the written paragraph (page 12, paragraph 1, line 519)

Comments 3: citations are missing in many places (e.g., lines 309-311: The KDIGO 2012 guidelines explained that interdisciplinary nephrological care should take into account patient education about dietary advice, renal replacement modalities, and transplantation

Response 3: Agree, we've added the KDIGO 2012 citation to page 11, paragraph 2 of point 4.1. Background, lines 488-490.

Comments 4: Line 175-177: It could become a drug of choice in CKD, compared to steroidal mineralocorticoid receptor antagonists [(?]).

Response 4: Agree, we have removed the sentence on page 5, paragraph: 2.1.1.2. Mineralocorticoid Receptor Antagonists.

Comments 5 : The abstract presents an interdisciplinary team consisting of a medical doctor, nurse, and social worker. The roles of nurse and social worker are not articulated in the paper, especially in the context of pharmacological and dietary support for patients with CKD

Response 5: We have clarified the points of discussion in this review. Page 3, paragraph 3, lines 92-94

Comments 6: Moreover, in lines 311-312, it states, “This team should consist of nephrologists, nephrology nurses, and renal dieticians;” thus, the social worker is not mentioned. It is not explained in any way.

Response 6: Agree, we have revised the sentence on page 11, section 4.1 Background, lines 491-492.

Comments 7: Chapter 2.1 is dedicated to Drug Nephroprotection, and its sub-chapters are focused on medications (2.1.1 - 2.1.3). So, why is Chapter 2.1.4 dedicated to Metabolic Acidosis and Chapter 2.1.5 to Anemia, while these are not drug categories.

Response 7: Agree, we have changed the sub-chapters on pages 4 to 7: Antihypertensive agents, Antidiabetic drugs, Sodium bicarbonate supplementation, Erythropoiesis-stimulating agents, Phosphorus Binders.

Comments 8: Chapter 2 is titled Drug and Non-drug Nephroprotection, so it is unclear why sub-chapter 2.3 is dedicated to Nephrotoxic Drugs.

Response 8: Agree, we have changed the subchapter, page 10, paragraph 3. Nephrotoxic Drugs, line 461

Comments 9: In Chapter 3 (The Interdisciplinary Team in Nephrology), there is a sub-chapter titled The Place of Clinical Pharmacist and Dietitians in CKD, but it contains only one paragraph, the middle one, focused on the dietician. This chapter seems to be the essence of the paper, but the roles of other team members are not adequately presented.

Response 9: We have added a section on the impact of the dietitian's interventions on page 14, paragraphs 2 and 3, lines 595-616.

Reviewer 2 Report

Comments and Suggestions for Authors

Dear Authors,

Thank you for the effort and dedication you have put into your manuscript, which effectively summarizes the findings of numerous studies. However, I feel there are several points that could be improved to enhance its clarity and impact.

While your article provides a broad overview, it differs little from many other reviews on similar topics. It remains unclear what specific benefits are derived from early pharmacological and dietary support for patients with moderate to severe (non-terminal) chronic kidney disease.

To provide greater depth, I would recommend including confidence intervals and data on risk reduction for each approach mentioned. For example, demonstrating that IRAS are particularly effective in reducing proteinuria, or illustrating that SGLT2 inhibitors do not lead to a long-term restoration of baseline eGFR, with dapagliflozin showing the weakest profile in this respect.

Perkovic V et al. Lancet Diabetes Endocrinol. 2018;6(9):691-704; Wanner C et al. N Engl J Med. 2016;375(4):323-34. Мosenzon O et al. Lancet Diabetes Endocrinol 2019; 7: 606–17 Zelniker TA et al. Online ahead of print. Lancet. 2018

Additionally, the role of GLP1 agonists,

Clin Kidney J, Volume 18, Issue 2, February 2025, sfae380, https://doi.org/10.1093/ckj/sfae380

which are currently in strong competition with SGLT2 inhibitors, should not be overlooked. Moreover, the role of non-steroidal mineralocorticoid receptor antagonists deserves attention.https://www.theisn.org/initiatives/toolkits/raasi-toolkit/#MRAs

I noticed the references primarily date up to 2023. Incorporating the latest research would add value and reflect a more comprehensive view of the field.

If your intention is to fully address the title of the article, I would recommend focusing on presenting robust evidence for each method, the sequence of their appropriateness in clinical application, and emphasizing that only those approaches contribute to the preservation of eGFR.

Nephrol Dial Transplant, Volume 40, Issue Supplement_1, January 2025, Pages i47–i58, https://doi.org/10.1093/ndt/gfae216

Alternatively, methods that hypothetically improve quality-of-life indicators, such as anemia treatment and mineral composition correction, should be positioned as secondary considerations. If such revisions are not feasible, I suggest reconsidering the scope and topic of the article to ensure clarity and alignment with its objectives.

I hope this feedback proves useful for improving your manuscript.

Best regards,

Author Response

Comments 1: To provide greater depth, I would recommend including confidence intervals and data on risk reduction for each approach mentioned. For example, demonstrating that IRAS are particularly effective in reducing proteinuria, or illustrating that SGLT2 inhibitors do not lead to a long-term restoration of baseline eGFR, with dapagliflozin showing the weakest profile in this respect.

Response 1: Thank you for pointing this out. We are agreeing with this comment. We have included confidence intervals on:

  • page 4, paragraph 2.1.1.1 Angiotensin-converting enzyme inhibitors and angiotensin II receptor antagonists, lines 142 - 152.
  • page 6, paragraph 2.1.2.1 Inhibitors of Sodium-Glucose Cotransport Type II, line 226
  • page 6, paragraph 2.1.2.2 Glucagon-like-peptide-1, lines 242- 244.
  • page 8, paragraph 2.2.2.1 Protein intake restriction, paragraph 1 and 2, lines 345- 349 and 359-362
  • page 8, paragraph 2.2.2.2 Sodium chloride intake restriction, paragraph 1, lines 367-370 and 375-380
  • page 9, paragraph 2.2.2.3 Potassium intake restriction, lines 392-407
  • page 9, paragraph 2.2.2.4, Phosphorus intake restriction, lines 442-444

But also, that dapagliflozin showing the weakest profile in this respect on page 6, paragraph 2.1.2.1   Inhibitors of Sodium-Glucose Cotransport Type II, on the line 228-229.

Comments 2 : Additionally, the role of GLP1 agonists,

Response 2: We've added a paragraph on GLP1 agonists on page 6-7, paragraph 2.1.2.2   Glucagon-like-peptide-1, lines 235-263

Comments 3: Moreover, the role of non-steroidal mineralocorticoid receptor antagonists deserves

Response 3: I have taken into account your article suggestion on page 5-6, paragraph 2.1.1.2. Mineralocorticoid Receptor Antagonists

Comments 4: I noticed the references primarily date up to 2023. Incorporating the latest research would add value and reflect a more comprehensive view of the field.

Response 4: We have added more recent research for GLP-1 and Mineralocorticoid Receptor Antagonists on page 6, paragraph 2.1.2.2 Glucagon-like-peptide-1 and 2.1.1.2.  Mineralocorticoid Receptor Antagonists

  • Perkovic, V.; Tuttle, K.R.; Rossing, P.; Mahaffey, K.W.; Mann, J.F.E.; Bakris, G.; Baeres, F.M.M.; Idorn, T.; Bosch-Traberg, H.; Lausvig, N.L.; et al. Effects of Semaglutide on Chronic Kidney Disease in Patients with Type 2 Diabetes. New England Journal of Medicine 2024, 391, 109–121,
  • Taal, M.W.; Selby, N.M. Glucagon-like Peptide-1 Receptor Agonists: New Evidence of Kidney and Cardiovascular Protection From the FLOW and SELECT Trials. American Journal of Kidney Diseases 2024, 85, 115–118, doi:10.1053/j.ajkd.2024.08.002.
  • Liabeuf, S.; Minutolo, R.; Floege, J.; Zoccali, C. The Use of SGLT2 Inhibitors and GLP-1 Receptor Agonists in Older Patients: A Debate on Approaches in CKD and Non-CKD Populations. Clin Kidney J 2025, 18, 380, doi:10.1093/CKJ/SFAE380.
  • Theodorakopoulou, M.; Ortiz, A.; Fernandez-Fernandez, B.; Kanbay, M.; Minutolo, R.; Sarafidis, P.A. Guidelines for the Management of Hypertension in CKD Patients: Where Do We Stand in 2024? Clin Kidney J 2024, 17, 36–50, doi:10.1093/ckj/sfae278.

Comments 5 : If your intention is to fully address the title of the article, I would recommend focusing on presenting robust evidence for each method, the sequence of their appropriateness in clinical applications, and emphasizing that only those approaches contribute to the preservation of eGFR.

Response 5: We have added an explanation regarding which part the critical review will focus on. (page 3, paragraph 3, lines 90-94)

Comments 6: Alternatively, methods that hypothetically improve quality-of-life indicators, such as anemia treatment and mineral composition correction, should be positioned as secondary considerations.

Response 6: We have considered your request on page 4, paragraph 2.1. Drug Renoprotection, lines 134-136

Reviewer 3 Report

Comments and Suggestions for Authors

The manuscript entitled “Benefit from Early Pharmacological and Dietary Support for Patients with Moderate to Severe (Non Terminal) Chronic Kidney Disease” is reviewed. The following comments may be considered:
1.     This review manuscript lacks novelty in the context of the contemporary advancement of this area of research. The author should include the novelty point of this review in the abstract and introduction section, as compared to the existing literature  (www.ncbi.nlm.nih.gov/pmc/articles/PMC5409713;  https://www.sciencedirect.com/science/article/pii/S2214623724000255; https://www.ajkd.org/article/S0272-6386(20)30726-5/fulltext).
2.    Authors should include some information on Clinical Trials and meta-analysis information on how Patients with Moderate to Severe CKD are affected and benefits of the use of dietary support.
3.    The author also should focus on CKD-induced other secondary complications (namely, CKD-induced metabolic bone loss, and anemia.
4.    Authors should also highlight the pathological changes that affect during early events of CKD and how Pharmacological and Dietary Support alleviate it.

Comments on the Quality of English Language

Language improved and typos may be proofread.

Author Response

Thank you for your comments. We have taken into account the need for an English Language revision of our article. We are awaiting feedback from all reviewers before submitting it for correction in English.

Comments 1: This review manuscript lacks novelty in the context of the contemporary advancement of this area of research. The author should include the novelty point of this review in the abstract and introduction section, as compared to the existing literature

Response 1: We agree, we have changed the title, page 1, line 1.

We have taken your sources into account except for one which we do not find to be relatively recent. (2017) (www.ncbi.nlm.nih.gov/pmc/articles/PMC5409713), we have supplemented with other more recent articles for the points :

  • 2.2.1 Protein intake restriction, paragraph 2, lines 359-362;
  • 2.2.3 Potassium intake restriction, paragraph 4, page 9-10, lines 423-427;
  • 2.2.5 Hydration Recommendations, page 10, lines 446-460.

We found this article ( https://www.sciencedirect.com/science/article/pii/S2214623724000255)

 interesting and have incorporated it into various points :

  • 2.1.4. Erythropoiesis-stimulating agents (page 7, line 279),
  • 2.2.3 . Potassium intake restriction (page 9, line 384),
  • 2.2.5 Hydration Recommendations (page 10, line 446),
  • 2 Place of Clinical Pharmacists and Dietitians in early CKD (page 12, line 541).

We have taken into account KDOQI 2020 as well as the new recommendations from KDIGO 2024 (reference number 1)

Comments 2: Authors should include some information on Clinical Trials and meta-analysis information on how Patients with Moderate to Severe CKD are affected and benefits of the use of dietary support.

Response 2: We agree. We have included meta-analyses and clinical trials at different points.

(reference:  3,13, 14, 15, 20, 23, 26, 33, 38, 43, 53, 55, 56,57, 76,85,89,90,93)

The role of the dietitian has been reworked in point 4.2. Place of Clinical Pharmacists and Dietitians in early CKD (page 14, paragraphs 2-3) Also, the point 2.2.2 Dietary Strategies (page 8, lines 337) was more detailed:

  • 2.2.1 Protein intake restriction, lines 345-349 and 359-362, page 8
  • 2.2.2. Sodium chloride intake restriction , lines 367-380, pages 8-9
  • 2.2.3. Potassium intake restriction, lines 390-409, 421-427 pages 9-10
  • 2.2.4. Phosphorus intake restriction lines 442-444, page 10
  • 2.2.5 Hydration Recommendations, lines 446-460, page 10

Comments 3: The author also should focus on CKD-induced other secondary complications (namely, CKD-induced metabolic bone loss, and anemia)

Response 3: We take your point into account. However, another reviewer advised discussing medications and not complications related to chronic kidney disease. However, we do discuss anemia in point 2.1.4 Erythropoiesis-stimulating agents, page 7, line 279.

Comments 4: Authors should also highlight the pathological changes that affect during early events of CKD and how Pharmacological and Dietary Support alleviate it.

Response 4: We have included a paragraph explaining the pathological changes in the stage, but we have not combined the explanation of pharmacological and dietary support together, but separately. On page 3, lines 90-94 and on the figure, page 4, line 130.

Round 2

Reviewer 1 Report

Comments and Suggestions for Authors

Line 474: The phrase 'pharmacological drug' doesn't make sense — the word 'pharmacological' should be removed.

No other concerns.

Reviewer 2 Report

Comments and Suggestions for Authors

Thank you for editing

Reviewer 3 Report

Comments and Suggestions for Authors

The authors have improved the manuscript. The English grammars and typos may be proofread.